

# Model spread in multidecadal NAO variability connected to stratosphere-troposphere coupling

Rémy Bonnet[1,2], Christine M. McKenna[3], Amanda C. Maycock[3]

[1]Institut Pierre-Simon Laplace, Sorbonne Université/CNRS, Paris, France

[2]CECI, Université de Toulouse, CERFACS/CNRS, Toulouse, France

[3]School of Earth and Environment, University of Leeds, Leeds, UK

*Correspondence to*: Rémy Bonnet (bonnet@cerfacs.fr)

**Abstract.** The underestimation of multidecadal variability in the winter-time North Atlantic Oscillation (NAO) by global climate models remains poorly understood. Understanding the origins of this weak NAO variability is important for making model projections more reliable. Past studies have linked the weak multidecadal NAO variability in models to an underestimated atmospheric response to the Atlantic Multidecadal Variability (AMV). We investigate historical simulations from CMIP6 large ensemble models and find that most of the models do not reproduce observed multidecadal NAO variability,

as found in previous generations of climate models. We explore statistical relationships with physical drivers that may contribute to intermodel spread in NAO variability. There is a significant anti-correlation across models between the AMV-NAO coupling parameter and multidecadal NAO variability over the full historical period (r=-0.55, p<0.05). However, this relationship is relatively weak and becomes obscured when using a common period (1900-2010) and detrending the data in a consistent way with observations to enable a model-data comparison. This suggests that the representation of NAO-AMV

coupling contributes to a modest proportion of intermodel spread in multidecadal NAO variability, although the importance of this process for model spread could be underestimated given evidence of a systematically poor representation of the coupling in the models. We find a significant intermodel correlation between multidecadal NAO variability and multidecadal stratospheric polar vortex variability and a stratosphere-troposphere coupling parameter, which quantifies the relationship between stratospheric winds and the NAO. The models with the lowest NAO variance are associated with weaker polar vortex

variability and a weaker stratosphere-troposphere coupling parameter. The two stratospheric indices are uncorrelated across models and together give a pooled $R^2$ with NAO variability of 0.7, which is larger than the fraction of intermodel spread related to the AMV ($R^2=0.3$). The identification of this relationship suggests that modelled spread in multidecadal NAO variability has the potential to be reduced by improved knowledge of observed multidecadal stratospheric variability; however, observational records are currently too short to give a robust constraint on these indices.



## 1 Introduction

The North Atlantic Oscillation (NAO) is the dominant mode of atmospheric circulation variability in the North Atlantic sector in winter and exerts a strong influence on regional weather and climate, especially over Europe and the US (Hurrell et al., 2003). A positive NAO phase is associated with a stronger meridional pressure gradient between the North Atlantic subtropical anticyclone and the Icelandic low, leading to a stronger North Atlantic eddy-driven jet stream and a northward displacement of the storm track. In winter, this brings mild and wet weather to northern Europe, and cold and dry weather to southern Europe.

Projections of wintertime surface climate over Europe depend on reliable simulation of the NAO (e.g., McKenna and Maycock, 2022). Several studies have shown that coupled climate models underestimate decadal to multidecadal NAO variability (e.g., Bracegirdle, 2022; Kravtsov 2017; Wang et al. 2017; Kim et al. 2018; Eade et al. 2022) and North Atlantic jet strength variability (Bracegirdle et al. 2018; Simpson et al. 2018). It has been proposed that the too low multidecadal North Atlantic atmospheric variability is related to simulated North Atlantic sea surface temperature (SST) variations and an underestimation by models of the atmospheric response to SST variability through too weak air-sea coupling (Kim et al. 2018; Bracegirdle 2022; Simpson et al. 2018).

Another important mechanism that influences winter NAO variability is the stratospheric polar vortex strength and the related coupling between the stratosphere and troposphere. On average, a weaker polar vortex leads to a negative winter NAO anomaly, and vice versa (Baldwin and Dunkerton, 2001). Low frequency polar vortex variability has been implicated in decadal surface climate trends (Garfinkel et al., 2018; Kretschmer et al., 2018; Zhang et al., 2016, Butler et al., 2023). A recent study from Zhao et al. (2022) highlighted a bias towards a weak polar vortex in the lower stratosphere for most of the CMIP6 climate models. Those models also have a large diversity in the representation of intraseasonal and interannual variability in the polar vortex (Charlton-Perez et al., 2013; Hall et al., 2021). However, the characteristics of multidecadal polar vortex variability in climate models are relatively understudied, in part because there are poor observational constraints.

The global tropics are also an important driver of the winter NAO, with myriad teleconnections emerging on sub-seasonal to seasonal (S2S) timescales from modes like ENSO (Ineson and Scaife, 2009), the MJO (Cassou, 2008) and IOD (Hardiman et al., 2020), and on decadal timescales from Interdecadal Pacific Variability (IPV) (Hu and Guan, 2018; Seabrook et al., 2023). While it has been suggested that tropical Pacific teleconnections to the NAO are too weak on seasonal timescales (Williams et al., 2023), the role for tropical forcing of the NAO on multidecadal timescales, and the extent to which this may contribute to the underestimated NAO variability in models, remains unclear.





Recent work has suggested a relationship between the NAO response to external drivers and the modelled relationship between eddy momentum fluxes and the extratropical zonal wind, a so-called 'eddy feedback parameter' (e.g. Smith et al., 2022; Hardiman et al., 2022; Screen et al., 2022). However, the extent to which eddy mean flow interactions may be linked to poor simulation of the large-scale circulation on multidecadal timescales is unclear.

Understanding the origin of the weak multidecadal NAO variability in models is important for resolving biases in climate models and making projections more reliable. In this paper, we further investigate the underestimation of the winter NAO multidecadal variability within the CMIP6 models by testing the extent to which the aforementioned mechanisms can explain the spread across climate models in their simulated NAO multidecadal variability.

The paper is organized as follows. The datasets, climate indices, and statistical methods used are described in section 2. The multidecadal NAO variability within the CMIP6 multi-model ensemble is then evaluated in Section 3. Section 4 analyses the origin of the spread in multidecadal NAO variability across the CMIP6 multi-model ensemble, highlighting the role of the polar vortex strength variability. Then, the origin of the spread in polar vortex strength variability within the CMIP6 multi-model ensemble is investigated. Finally, the main limitations of this study are discussed, and conclusions and perspectives are drawn in Section 5.

## 2 Data and Methods

### 2.1 Datasets

The historical simulations from 15 CMIP6 models (Eyring et al., 2016) are used in this study (Table 1). To analyse drivers of low frequency climate variability requires long simulations to find physical relationships and reduce the likelihood of spurious correlations due to poor sampling. Therefore, we analyse CMIP6 models providing at least 10 ensemble members for the DECK historical experiment with daily zonal wind (ua) and meridional wind (va) variables available that are needed to calculate the eddy feedback parameter described in Section 1 (see Section 2.2.5). All atmospheric data are regridded to the horizontal resolution of CanESM5, which is the coarsest grid, using bilinear interpolation. The SST data are regridded over a regular 1°x1° grid using bilinear interpolation.

| Model | Number of simulations | Model | Number of simulations |
|---|---|---|---|
| ACCESS-ESM1-5 | 40; r[1-40]i1p1f1 | IPSL-CM6A-LR | 33; r[1-33]i1p1f1 |
| CanESM5 | 35; r[1-25]i1p1f1 and r[1-10]i1p2f1 | MIROC-ES2L | 30; r[1-30]i1p1f2 |
| CESM2 | 11; r[1-11]i1p1f1 | MIROC6 | 10; r[1-10]i1p1f1 |



| CMCC-CM2-SR5 | 11; r[1-11]i1p1f1 | MPI-ESM1-2-HR | 10; r[1-10]i1p1f1 |
|---|---|---|---|
| CNRM-CM6-1 | 30; r[1-30]i1p1f2 | MPI-ESM1-2-LR | 30; r[1-30]i1p1f1 |
| CNRM-ESM2-1 | 10; r[1-10]i1p1f2 | MRI-ESM2-0 | 10; r[1-10]i1p1f1 |
| EC-Earth3 | 23; r[1-25]i1p1f1; no r5 and r8 | UKESM1-0-LL | 17; r[1-19]i1p1f2; no r13 and r14 |
| INM-CM5-0 | 10; r[1-10]i1p1f1 | | |

**Table 1: Summary of the 15 CMIP6 models and the associated number and list of DECK historical simulations used in this study.**

90

Since our focus is on multidecadal variability, to estimate observed NAO variability we use two long-term atmospheric reanalyses, the NOAA-CIRES-DOE 20th Century Reanalysis version 3 (20CRv3; Slivinski et al., 2019) and the ECMWF 20th Century Reanalysis (ERA20C; Poli et al., 2016), as well as the Hadley Centre Sea Level Pressure dataset (HadSLP2r; Allan and Ansell, 2006). Two long-term observational datasets of SST are also used: the Hadley Centre Sea Ice and Sea Surface

95    Temperature data set (HadISST; Rayner et al., 2003) and the NOAA Extended Reconstructed SST V5 (ERSSTv5; Huang et al., 2017). We analyse the observation-based datasets over their common period 1900-2010. It is important to recognise that the characterisation of low frequency variability in instrumental datasets is rather limited due to the low degrees of freedom. Therefore, when evaluating the large ensemble model simulations, we ask where the observations lie within the ensemble spread to determine the likelihood that a model is biased (Maher et al. 2021).

100

The analysis focuses on the extended December to March winter period, since several recent studies point out that the underestimation of North Atlantic atmospheric variability, including the NAO, is also present in March (Simpson et al., 2018; Bracegirdle, 2022).

**2.2 Climate indices**

105    **2.2.1 NAO index**

Following Stephenson et al. (2006) and Baker et al. (2018), the NAO index is defined as the difference in area-averaged mean sea level pressure (MSLP) between a southern box (90°W–60°E, 20°N–55°N) and a northern box (90°W–60°E, 55°N–90°N) in the North Atlantic. We choose this index because it is less sensitive to modest differences in NAO centers of action between the observations and the CMIP6 models than the station-based index (Hurrell et al., 2003; Stephenson et al., 2006). Another

110    benefit of this index is that it is less affected by issues of interpretability that occur when using a mathematically constructed empirical orthogonal function (EOF)-based index (Ambaum et al., 2001; Dommenget and Latif, 2002; Stephenson et al., 2006).





### 2.2.2 AMV definition

The AMV is the leading mode of multidecadal variability in the North Atlantic Ocean and is characterized by basin wide SST variations (Schlesinger and Ramankutty, 1994; Enfield et al., 2001; Yeager and Robson, 2017). To estimate the evolution of the AMV, the AMV index is generally defined as the average SST over the North Atlantic (0-60°N, 80°W-0°E) after the removal of the externally forced signal. A low-pass filter is then used to retain only the low-frequency variations. The most accurate way to estimate the external forcing from climate simulations is to use the ensemble mean when enough simulations are available (Deser et al., 2020). However, this is not possible for the observations. Therefore, we also use the Trenberth and Shea (2006) method (TS2006 hereafter), which estimates the effect of external forcings by removing the global averaged SST between 60°S and 60°N from North Atlantic SST, although other methods can be used (Qasmi et al., 2017).

### 2.2.3 Interdecadal Pacific Oscillation definition

The Interdecadal Pacific Oscillation (IPO) is characterized by a horseshoe pattern of SST variability over the North Pacific. A positive phase of the IPO is associated with an eastern warming and a cooling in the Kuroshio-Oyashio Extension, similar to the Pacific Decadal Variability pattern, a warming over the tropical Pacific region, and a cooling over the southwestern Pacific Ocean (Newman et al., 2016). We use the tripole TPI IPO index (Henley et al., 2015), which is the weighted difference between deseasonalised monthly SST anomalies (SSTA) over the central equatorial Pacific (SSTA$_2$, 10°S–10°N, 170°E−90°W), the Northwest (SSTA$_1$, 25°N–45°N, 140°E–145°W) and the Southwest Pacific (SSTA$_3$, 50°S–15°S, 150°E–160°W): TPI $=$ SSTA$_2$−(SSTA$_1$−SSTA$_3$)/2.

### 2.2.4 Polar vortex strength

To estimate the multidecadal variability of the stratospheric polar vortex, we calculate the variance of the 20-year mean extended winter (DJFM) zonal mean zonal wind averaged between 60-70°N at 50 hPa (Castanheira and Graf, 2003; Walter and Graf, 2005).

Sudden stratospheric warmings (SSWs) are a key feature of the northern hemisphere polar vortex during which the vortex rapidly breaks down and typically recovers over a period of weeks to months. Past work has identified multidecadal variability in SSW frequency (Dimdore-Miles et al., 2022). To identify SSWs, we use the index of Charlton and Polvani (2007) based on the temporary reversal of zonal mean zonal wind at 60°N and 10 hPa. To be considered as discrete SSW events, periods of wind reversal to easterly must be separated by at least 20 consecutive days of westerly winds. Only SSWs with an onset date between December and March are included. We calculate the 20-year mean winter SSW frequency and examine whether multidecadal variability in winter polar vortex strength is related to SSW frequency (e.g., Jucker et al., 2014).



### 2.2.5 Eddy feedback parameter

To quantify the so-called eddy feedback parameter (EFP) we follow ⬚Hardiman et al. (2022)⬚. From the daily zonal (u) and meridional (v) winds at 500 of the horizontal EP-flux divergence ⬚⬚(Andrews et al., 1987):

$$\frac{\nabla.F_h}{\rho a \cos\phi} = \frac{-1}{a \cos^2\phi}\frac{\overline{du'v'}\cos^2\phi}{d\phi}$$

where ρ is density, φ is latitude, a is Earth's radius, overbars represent a zonal mean, and primes represent the residual after removing the zonal mean. The extended winter mean is then calculated from the daily values of this zonal acceleration. In parallel, the extended winter mean of the zonal mean zonal wind ($\bar{u}$) is computed for each year. Next, the time correlation is
calculated at each latitude between the zonal acceleration and the zonal mean zonal wind ($\bar{u}$). Finally, the eddy feedback parameter is calculated as the area-weighted average of this correlation squared over 25°N–72°N. It is important to note the EFP does not formally represent the feedback of eddies onto the mean flow (e.g. Lorenz and Hartmann, 2001), rather it reflects the cross-correlation between EP-flux divergence and zonal wind.

### 2.3 Statistical methods

Our analysis focuses on explaining the intermodel spread of multidecadal NAO variability in the CMIP6 multi-model ensemble. To estimate the error of the ensemble mean variance (μ) and the possibly related variables for each CMIP6 model, two-sided confidence intervals are calculated as $\mu \pm \sigma/N$, where $\sigma$ is the standard deviation across the N ensemble members (Storch and Zwiers, 1999). To test the significance of the relationship between two variables, the p-value is provided from a two-tailed Wald Test with a null hypothesis that the slope is zero.

## 3 Evaluation of simulated winter NAO variability

We first evaluate the representation of historical multidecadal winter NAO variability in the CMIP6 large ensembles compared to the observation-based datasets. Figure 1 shows the winter NAO index for the three observation-based MSLP datasets. While there are similar inter-decadal variations in the datasets, there are discrepancies in the apparent long-term trend. There is almost no long-term trend in HadSLP2r over this period, whereas there is a positive linear trend of 1.49 and 1.38 hPa/century in the
ERA20C and the 20CRv3 reanalyses, respectively. This trend leads to larger multidecadal variability in those datasets as compared to HadSLP2r (Fig. 1a). Studies have highlighted potential unrealistic trends in long-term reanalyses (Krueger et al., 2013; Oliver, 2016) because only a limited set of surface observations are assimilated (sea level pressure and surface wind for ERA20C) and the density of the observation network evolves with time. HadSLP2r is based on station observations whose density also changes through time. Therefore, given the differences in composition of the datasets, it is difficult to assess the





validity of their long-term NAO trends. Detrending the datasets results in a closer evolution of the multidecadal NAO and the

associated multidecadal variance (Fig. 1b). On multidecadal timescales, an apparent long-term trend may reflect externally-

forced changes, but may also be affected by the phasing of internal variability relative to the trend end points. Therefore,

detrending may remove some of the unforced variability we are interested in. Nevertheless, given the differences amongst

datasets, unless otherwise stated the remaining analyses for observations and models use linearly detrended timeseries for all

variables.

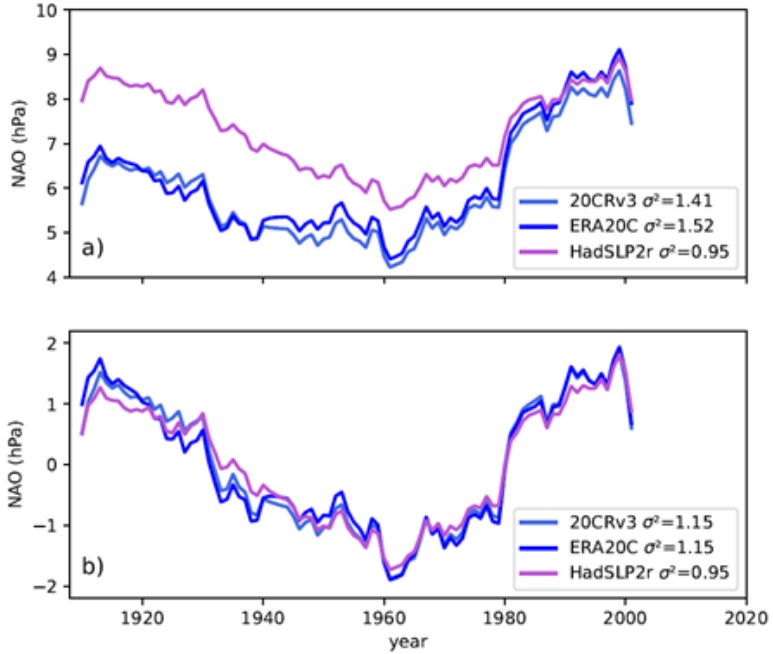

**Figure 1: (a) Evolution of the extended winter (DJFM) NAO for the 20CRv3 reanalysis, the ERA20C reanalysis and for the HadSLP2r Sea Level Pressure dataset over their common period (1900-2010). (b) Same as (a) but with the 1900-2010 linear trend removed. A running mean with a 20-year window is applied to each dataset. The variance $\sigma^2$**

**calculated over the whole period for each dataset is given in the legend.**

The variance of the 20-yr running mean NAO in the observation-based datasets lies within the extreme upper range of the

CMIP6 model distributions, with only a few realisations having variance above or close to that observed (Fig. 2). As the

variance of the HadSLP2r dataset is 17% lower than the two reanalyses, this dataset is somewhat more consistent with the

CMIP6 simulations, although it is still within the very upper range. Most of the models have no simulations close to the

observed variance. This underestimation is still visible but lower when using a 10-yr running mean NAO for the ERA20C and

20CRv3 reanalyses (Supplementary Fig. S1). For HadSLP2r, the underestimation is even smaller and is not visible for some

models, although some of them still fail to reproduce the variability. This quasi-systematic underestimation of the NAO



variance by CMIP6 models was also highlighted in recent studies (e.g. Schurer et al., 2023). The fact that very few simulations
are able to reproduce the observed variability suggests that a bias seems to be present in climate models. However, it is possible,
albeit unlikely, that the observations are characterised by a very high low-frequency internal variability by chance.

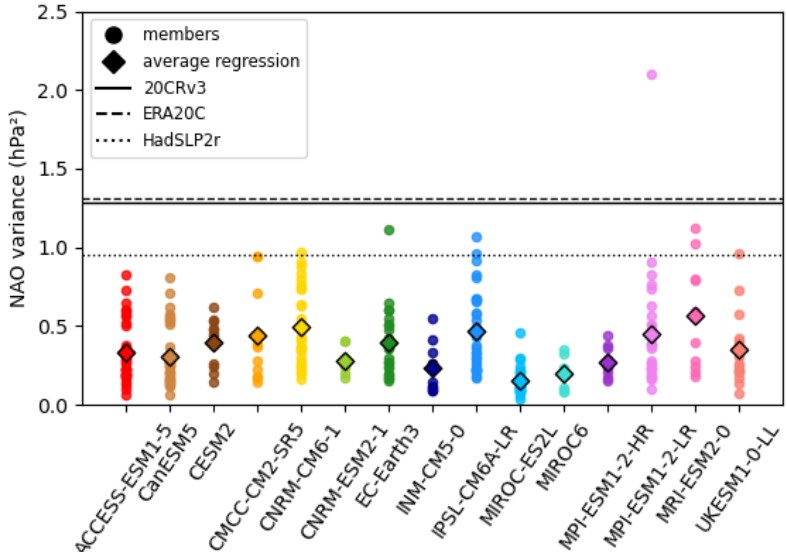

**Figure 2: December to March detrended 20-yr running mean NAO variance (hPa²) for each member of the 15 CMIP6**
**ensembles (dot), the ensemble mean (diamond) and for the three observation-based datasets: 20CRv3 (solid line),**
**ERA20C (dashed line) and HadSLP2r (dotted line) calculated over the 1900-2010 period.**

While the systematic underestimation of multidecadal NAO variance compared to observations is clear in Fig. 2, there are also
differences in variance between individual models. The ensemble mean low frequency NAO variance in Fig. 2 varies by up to
around a factor of three, from 0.15 to 0.57 hPa². In order to identify some of the factors that may influence the representation
of low frequency NAO variance in models, for the remainder of the study we focus on exploring the intermodel spread in
ensemble mean NAO variance and its relationship to other climate parameters. For consistency with observations, we focus
on the 1900-2010 period and use detrended time-series of climate parameters.



# 4 Origins of the intermodel spread in multidecadal winter NAO variability in CMIP6

## 4.1 Relationship of NAO variability with Atlantic Multidecadal Variability

We first evaluate the relationship amongst models between low frequency NAO variance and the simulated AMV. It has been shown that a negative NAO phase follows a positive AMV phase in both observations (Peings and Magnusdottir, 2014; Gastineau and Frankignoul, 2015) and simulations from CMIP5 models (Gastineau et al., 2013; Peings et al., 2016) as well as some CMIP6 models (Ruggieri et al., 2021; Börgel et al., 2022). These findings suggest a positive feedback between the AMV

and the NAO, with positive SST anomalies in the North Atlantic leading to a negative NAO phase that subsequently reinforces the positive AMV, and vice versa. Such coupled feedbacks would enhance the total low-frequency NAO variability (e.g. Farneti and Vallis, 2011). Therefore, an underestimation of AMV variability and/or the coupling between the AMV and the overlying atmospheric circulation could introduce deficiencies in the NAO variability (Bracegirdle, 2022).

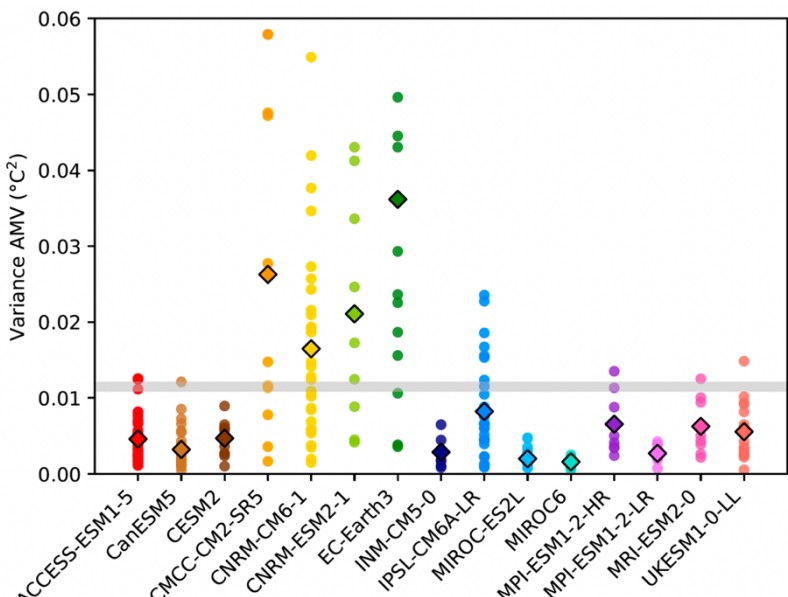


**Figure 3: December to March 20-yr running mean AMV variance (°C²) for individual members (dots), the ensemble means (diamond) and two observational datasets, HadISST and ERSSTv5 (shaded gray bar) calculated over 1900-2010.**

There is substantial diversity in simulated AMV in the CMIP6 models, with different magnitudes (Fig. 3) and periodicities (Supplementary Fig. S2). Some models are characterized by larger ensemble mean AMV variability and larger ensemble spread which encompasses the observations (CMCC-CM2-SR5, CNRM-CM6-1, CNRM-ESM2-1, EC-Earth3 and IPSL-CM6A-LR), while others have very weak average AMV variability and smaller ensemble spread (INM-CM5-0, MIROC-




ES2L, MIROC6, MPI-ESM1-2-LR). Despite the spread in AMV representation across models, no significant relationship is
found between the ensemble mean variance of the NAO and the AMV over the 1900-2010 period (Fig. 4). This is also the case
when using 10 or 30 year running means, as well as considering the whole historical period 1850-2014 (not shown).

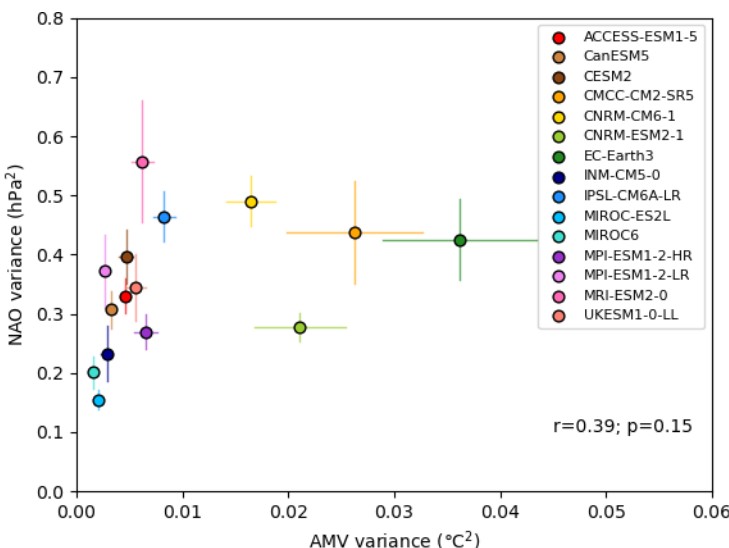

**Figure 4: Scatter plot of the ensemble mean 20-yr running mean NAO variance (hPa²) versus 20-year running mean**
**AMV index variance (°C²) in each model for DJFM over 1900-2010.**

While there isn't a relationship between the total NAO and AMV variances across models, there may be differences in the
coupling parameter between the multidecadal NAO and AMV. We evaluate the coupling between the AMV and the NAO
using the linear regression slope between the 20-year running mean NAO and AMV timeseries in each ensemble member (Fig.
5). A large range of apparent NAO-AMV coupling amplitudes are found in individual members, with simulations with a strong
positive relationship and others with a strong negative relationship, even within the same climate model (Fig. 5a). The observed
NAO-AMV regression slope is toward the lower end of the range of all the simulations. This could either mean the atmosphere-
ocean coupling is biased in the models, or that the observations by chance exhibit a strong negative relationship between the
AMV and the NAO. Figure 5a reflects the instantaneous relationship between the NAO and AMV; when the AMV leads the
NAO the underestimation of the relationship compared to observations is even larger (Fig. 5b). Conversely, when the NAO
leads the AMV, the observed coupling between the AMV and the NAO is closer to the simulated range. This suggests that the
atmosphere forcing the ocean may be better simulated than the ocean forcing the atmosphere.



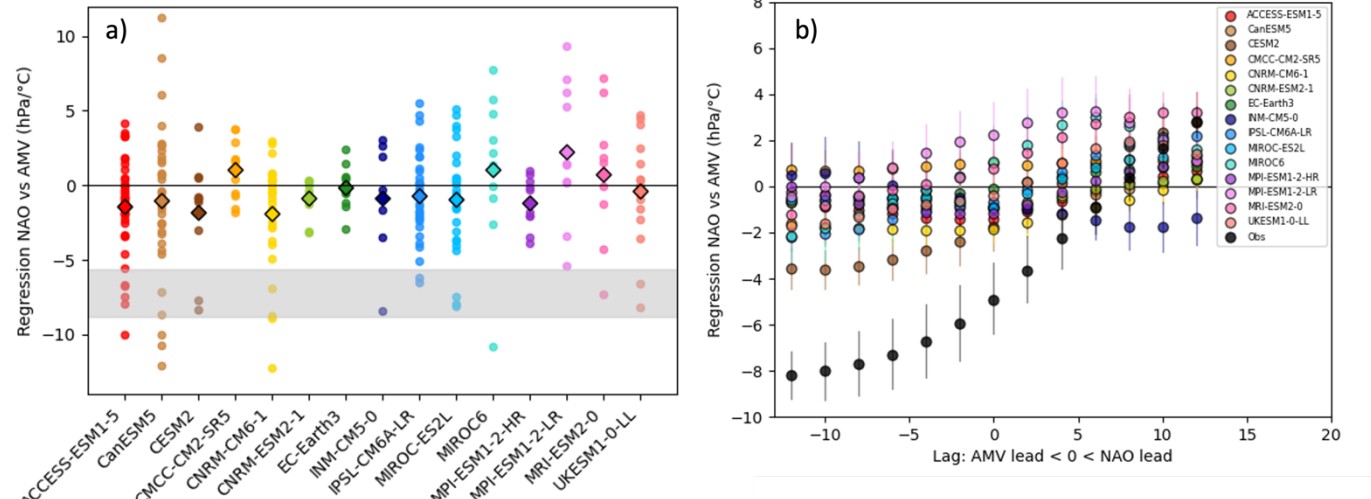

**Figure 5: (a) Regression slope between the 20-year running mean NAO and AMV indices for DJFM over 1900-2010 for each model ensemble member (dot) and the ensemble mean (diamond). The observed range of the regression slope (grey area) is defined as the minimum and the maximum of the slopes calculated from all permutations of the observational datasets (HadISST and ERSSTv5 for the AMV and 20CRv3, ERA20C and HadSLP2r for the NAO). (b) Lead-lag regression slopes of the relationship between the AMV and the NAO for each model and the observations**
**(dot) and their relative uncertainties (bars).**

To analyze if the intermodel spread in the AMV-NAO coupling parameter is related to the spread in low frequency NAO variance, we use the regression slope with the AMV leading the NAO by 10 years, in which a large fraction of the models have an ensemble mean negative relationship qualitatively consistent with observations (Fig. 5b). This appears to show there

is no significant relationship between the multidecadal NAO variability and the AMV-NAO coupling across models (Fig. 6). However, if we use the full historical period (1850-2014) and remove the forced component of the NAO and the AMV indices by subtracting the ensemble mean timeseries, we do find a significant anti-correlation (Supplementary Fig. S3a; r=-0.55, p-value=0.04) that was obscured by adopting a consistent methodology with the observations. This relationship means models with a stronger negative NAO-AMV regression slope have larger low frequency NAO variability, in agreement with the

relationship seen in observations (Fig. 5a). We note the relationship is also stronger and more significant when using a 10-yr running mean instead of 20-yr (Supplementary Fig. S3b). While this study is focused on the intermodel spread, it is possible that all the models are systematically biased in the same way with respect to their atmosphere-ocean coupling, which could contribute to the systematic underestimation of low frequency NAO variance in almost all models (Fig. 2).




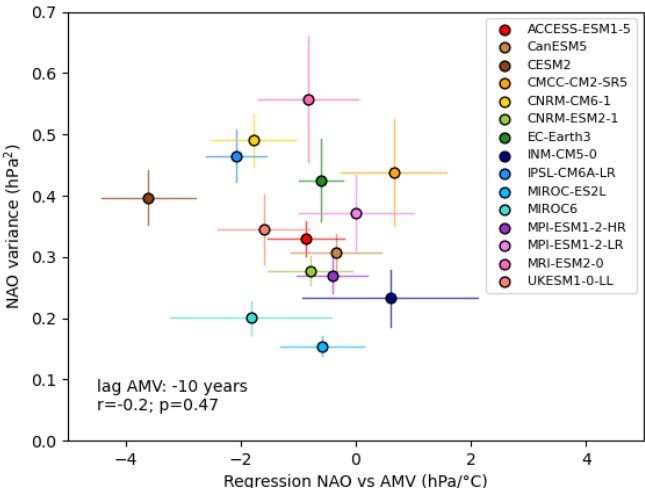


**Figure 6: Scatter plot of the ensemble mean 20-year running mean NAO variance (hPa²) versus the regression slope between the 20-year running mean NAO and AMV for DJFM over the 1900-2010. The AMV leads the NAO by 10 years.**

**4.2 Relationship of NAO variability with Interdecadal Pacific Variability**

In this section, we explore the possible role of multidecadal variability in Pacific SSTs to explain the spread in multidecadal NAO variability. Indeed, several studies suggest the existence of a teleconnection between low-frequency Pacific SST and the North Atlantic atmospheric variability (e.g. Smith et al., 2016; Weisheimer et al., 2017; Seabrook et al., 2023).

There is no significant relationship between the ensemble mean variance of the TPI IPO index and the low frequency NAO
variance (Fig. 7a). As for the AMV index, there is also no significant relationship across models between the NAO-IPO regression slope and the low frequency NAO variance (Fig. 7b). The large ensemble simulations demonstrate a substantial influence from internal variability on estimating the relationships within single realisations and potentially the observational record, due to the relatively small number of degrees of freedom when considering low frequency variability (Supplementary Fig. S4).





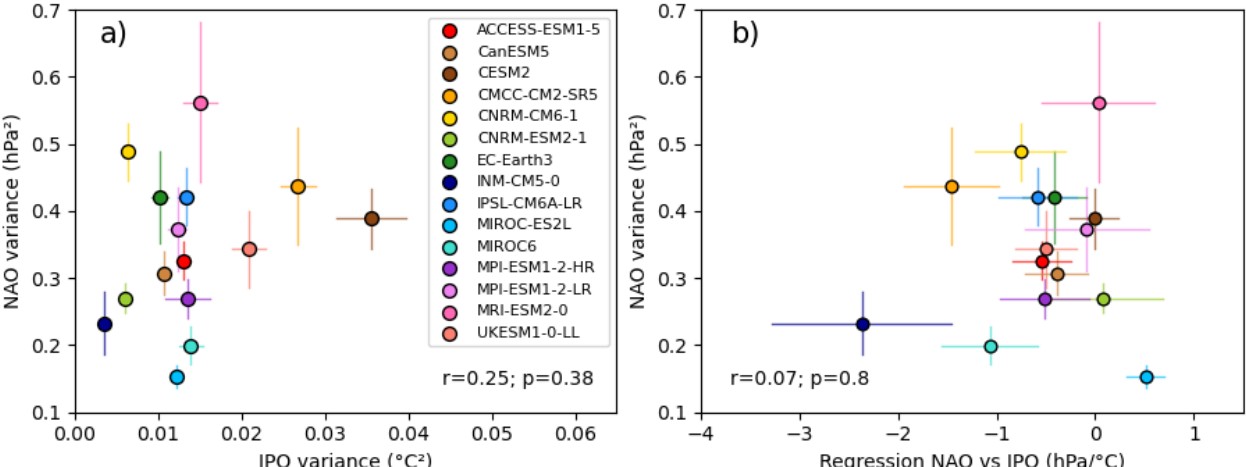

**Figure 7: (a) Scatter plot of the ensemble mean low frequency NAO variance (hPa²) versus (a) the TPI IPO index variance (°C²) and (b) the regression slope between the NAO and TPI IPO indices (hPa/°C) calculated for DJFM over 1900-2010.**

### 4.3 Relationship of NAO variability with stratospheric polar vortex

In this section, we explore the potential role of the polar vortex to explain the spread in low frequency NAO variability within the CMIP6 models. A causal link between the polar vortex strength and the NAO has been widely demonstrated at subseasonal-to-seasonal timescales (e.g. Baldwin et al., 1994; Baldwin and Dunkerton, 2001; Jung and Barkmeijer, 2006; Hitchcock and Simpson, 2014) and on multidecadal timescales (Scaife et al., 2005; Garfinkel et al., 2017; Kretschmer et al., 2018; Butler et al., 2023).

There is a significant positive relationship between low frequency NAO variability and low frequency polar vortex variability within the CMIP6 models with an $R^2 = 0.33$ (Fig. 8a). However, this relationship is sensitive to the inclusion of two outlier models with particularly low NAO and polar vortex variability (MIROC-ES2L and MIROC6) and when these models are removed the correlation across models is not significant. We next examine the stratosphere-troposphere coupling strength in the models, estimated as the regression of the low frequency NAO index onto the polar vortex strength (Fig. 8b; cf. Maycock and Hitchcock, 2015). The stratosphere-troposphere coupling parameter is positive, consistent with the wide literature showing a stronger vortex is coincident with an anomalously positive NAO index, and vice versa (e.g., Charlton and Polvani, 2007). The stratosphere-troposphere coupling parameter on multidecadal timescales is linearly correlated with the parameter estimated using interannual data across models (Supplementary Fig. S5). This is useful to note because while the satellite record is not yet long enough to constrain stratosphere-troposphere coupling on multidecadal timescales, it is possible to estimate this parameter on interannual timescales (Maycock and Hitchcock, 2015).



The vortex variability and stratosphere-troposphere coupling strength are not correlated with one another (r=0.38, p=0.17, Supplementary Fig. S6), indicating these are largely independent factors that relate to simulated low frequency NAO

variability. Therefore, both the intermodel spread in the polar vortex strength variance and the intermodel spread in the coupling between the NAO and the polar vortex can explain a large fraction of the spread in the NAO variance at multidecadal timescales. A multilinear regression model including both terms and accounting for their cross-correlation produces a combined $R^2$ of 0.72. In any single realisation, there is a large uncertainty in the low frequency NAO-vortex strength regression slope (Supplementary Fig. S7), likely due to the relatively low degrees of freedom, indicating the importance of using large

ensembles to investigate these multidecadal relationships.

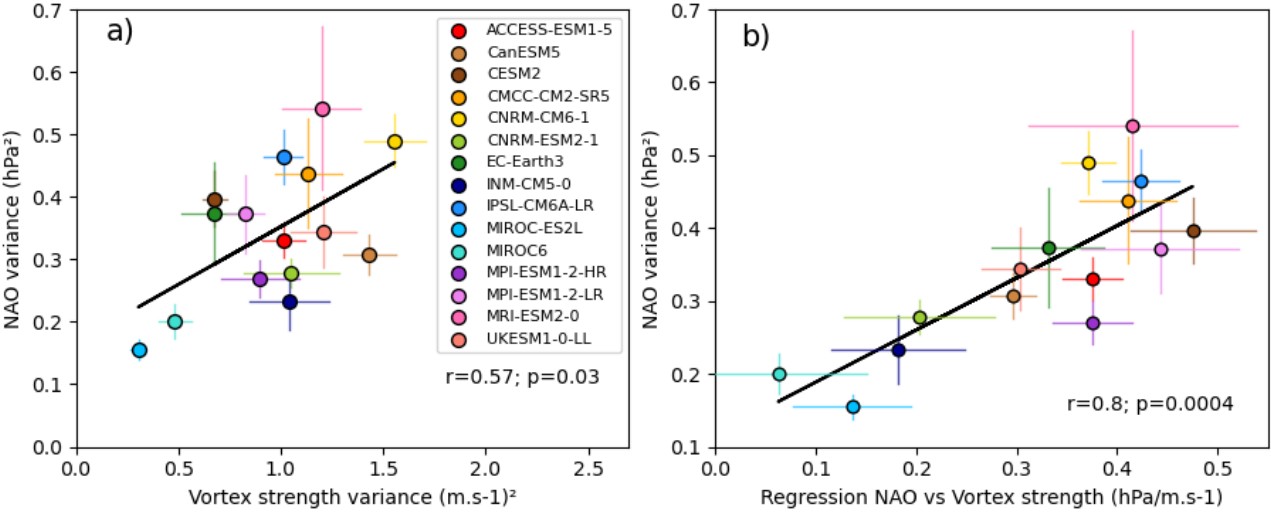

**Figure 8: Scatter plot of the ensemble mean low frequency NAO variance versus (a) the low frequency polar vortex variance (m/s)² and (b) the regression slope between polar vortex strength and the NAO (hPa/ms⁻¹) for DJFM over**

**1900-2010. The black line represents the least square regression with Pearson correlation and p-value (see Section 2.3).**

While we cannot isolate causality in these relationships, it has been shown that when a multidecadal stratospheric vortex trend is imposed in a model the NAO trend is affected (e.g., Scaife et al., 2005). On these timescales, variability in the vortex strength could be affected by external forcing (e.g., solar forcing, volcanoes) or by internally generated unpredictable chaotic variability.

While there is a known stratospheric pathway linking tropical Pacific SSTs to the NAO on seasonal timescales (Trascasa-Castro et al., 2019), the lack of relationship between the IPO and low frequency NAO suggests the Pacific is not a key driver of modelled NAO spread via a stratospheric pathway on multidecadal timescales.





## 5 Possible origins of the spread in polar vortex variability and NAO-polar vortex coupling

The previous section identified a relationship across models between the winter NAO multidecadal variability and the polar
vortex variance, as well as with the regression slope between the low frequency vortex strength and the NAO. This raises
questions about the origins of the spread across models in these polar vortex related parameters.

One candidate to explain the spread in the low frequency polar vortex variability is the representation of sudden stratospheric
warmings (SSW). Some studies suggest that models with higher vortex variability also have higher SSW frequency (Hall et al.
2021), whereas other authors have regarded SSWs as the tail of a more normally distributed spectrum of polar vortex variability
(Horan and Reichler, 2017). Here, we ask whether the spread in multidecadal vortex variability across CMIP6 models is related
to low frequency variability in the occurrence of SSWs. There is some hint in the reanalysis record of decadal variability in
SSW frequency (Domeisen et al. 2019). Note that daily zonal wind data from MRI-ESM2-0 and ACCESS-ESM1-5 are only
available from 1950. No significant relationship is found between the variability of 20-year running mean SSW frequency and
the low frequency polar vortex variability over 1900-2010 (Figure 9a). It is possible that low frequency variability in the upward
propagation of planetary wave activity would explain the vortex variability (e.g., Schimanke et al. 2011), but this is beyond the
scope of the current study.

We now investigate a potential explanation for the intermodel spread in stratosphere-troposphere coupling strength. Some
recent studies have identified a relationship between the amplitude of Northern hemisphere extratropical circulation responses
to external drivers and an estimate of the interaction between eddies and the mean flow (so-called eddy feedback parameter
(EFP), see Section 2.2.5). These studies have suggested that models with a weaker EFP exhibit weaker Northern hemisphere
extratropical circulation signals. The tropospheric response to polar vortex variability involves amplification of flow anomalies
by eddy feedbacks (e.g., Song and Robinson, 2004; Domeisen et al., 2013; Hitchcock and Simpson, 2016), so if eddy-mean
flow feedbacks were represented differently in models, this could contribute to spread in the stratosphere-troposphere coupling
strength.



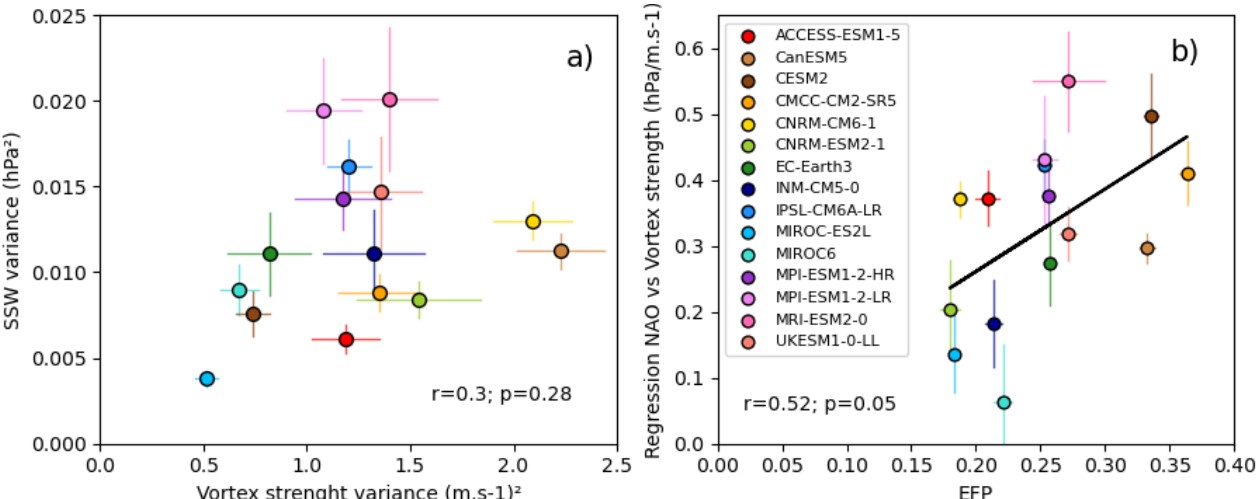

**Figure 9: a) Scatter plot of the ensemble mean 20-yr running mean SSW variance (hPa²) versus the polar vortex**
**variance (m.s⁻¹)² for DJFM over 1900-2014. b) Scatter plot of the ensemble mean of the regression slope between 20-yr**
**mean NAO and 20-yr mean polar vortex strength (hPa/m.s⁻¹) for DJFM over 1900-2014 versus the EFP calculated for**
**each of the ensemble of climate model simulations for the DJFM months over the 1850-2014 period. For MRI-ESM2-0**
**and ACCESS-ESM1-5 models, the SSW and EFP periods start in 1950, as they only have daily data available from**
**1950 onwards. The black line represents the least square regression with Pearson correlation and p-value (see Section**
**2.3).**

We calculate the EFP in the CMIP6 models using the full historical period for all the models (1850-2014) except for MRI-ESM2-0 and ACCESS-ESM1-5 as they only have daily data available from 1950 onwards. There is a significant positive relationship between the stratosphere-troposphere coupling parameter and the EFP with r=0.61 (p=0.02) (Fig. 9b). Models
with a higher EFP exhibit a stronger stratosphere-troposphere coupling parameter. Conversely, there is no relationship across models between the low frequency variability in polar vortex strength and the EFP (Supplementary Fig. S8), suggesting tropospheric eddy-mean flow interaction is unrelated to spread in polar vortex variability. The latter may be more related to the representation of planetary scale wave forcing and this would be an area for future study.

## 6. Discussion and conclusions

In this study, we investigated the representation of multidecadal NAO variability within the CMIP6 historical large ensemble models and explored statistical relationships with physical factors that could potentially explain intermodel differences in simulated multidecadal NAO variability. When using the full historical simulation (1850-2014) and detrending by removing the ensemble mean, we do find a significant intermodel relationship between the NAO-AMV regression parameter and



multidecadal NAO variability (r=-0.55, p<0.05). This suggests that intermodel spread in multidecadal NAO variability can be
partly explained by the representation of AMV-NAO coupling. However, the relationship between AMV-NAO coupling and
multidecadal NAO variability across models is quite weak and disappears when using a consistent methodology to that applied
in observations (i.e. using a common period with observations and linearly detrending indices). This overall weak role for the
AMV on the NAO may be related to weak atmosphere-ocean coupling in models as has been suggested in other studies (e.g.,
Simpson et al., 2018).


We examine other potential factors that may relate to multidecadal NAO variability and find that the representation of
Interdecadal Pacific Variability does not explain the spread in multidecadal NAO variability, which may arise through tropical-
extratropical teleconnections.  In contrast, there is a statistically significant relationship across models between multidecadal
variability in the stratospheric polar vortex strength and NAO variability (r=0.57, p<0.05). Furthermore, there is a significant
relationship across models between the magnitude of a multidecadal stratosphere-troposphere coupling parameter and
multidecadal NAO variability (r=0.8, p<0.05), which is largely independent of the vortex strength variability. Together these
two measures explain around 70% of the variance in multidecadal NAO variability across models, which is a larger proportion
of the intermodel spread than can be explained by the representation of the AMV. While similar relationships have been found
in other studies (e.g., Maycock and Hitchcock, 2015), these statistical relationships do not isolate causality. It is possible that
the NAO variability itself drives the polar vortex variability, or that both factors are related to a common, unidentified cause.
Nevertheless, there is a wide body of literature demonstrating a causal influence of the polar vortex on the NAO on
intraseasonal (Hitchcock and Simpson, 2014), interannual (Ineson and Scaife, 2009; Bell et al., 2009) and decadal timescales
(Scaife et al., 2005). Therefore, based on knowledge from the wider literature, we hypothesise that the representation of polar
vortex variability and the represented strength of stratosphere-troposphere coupling are both important factors for simulating
multidecadal NAO variability. Unfortunately, these parameters cannot be well estimated from observations because the record
of stratospheric data is too short to robustly assess multidecadal variability. Nevertheless, the stratosphere-troposphere
coupling parameter on multidecadal timescales is correlated with the parameter on interannual timescales in models. The
reanalysis record is long enough to estimate the parameters on interannual timescales, so that may offer a route to constraining
the model spread.


We find that the intermodel spread in the stratosphere-troposphere coupling parameter is correlated with the relationship
between eddy momentum forcing and the zonal mean flow in the extratropical Northern hemisphere troposphere. On average,
a weaker correlation between eddies and zonal mean flow anomalies coincides with a weaker stratosphere-troposphere
coupling parameter. This may be indicative of the recognised role of tropospheric eddy feedbacks in amplifying and
maintaining the tropospheric response to stratospheric anomalies (e.g., Song and Robinson, 2004; Domeisen et al., 2013).
Weak 'eddy feedback' has been hypothesised as a contributor to the too weak Arctic Oscillation predictability within climate



models on seasonal timescales (Hardiman et al., 2022). The apparent relationship between the stratosphere-troposphere coupling parameter and the eddy-mean flow coupling should be further investigated in controlled experiments.

## Data availability statement

All CMIP6 data are available through the Earth System Grid Federation. ERA20C is available from the C3S store. 20CRv3 and ERSSTv5 are provided by the NOAA Earth System Research Laboratory's Physical Sciences Division (PSD), Boulder, Colorado, USA, from their website: https://www.psl.noaa.gov/data/gridded/data.20thC_ReanV3.html. HadSLP2r and HadISST are provided by the Met Office Hadley Centre observations datasets from their website: https://www.metoffice.gov.uk/hadobs.


## Author contribution

R.B., C.M. and A.M. designed the study. R.B. and C.M performed the calculations of the indices and the analyses. RB prepared the manuscript with contributions from all co-authors.

## Competing interests

The authors declare that they have no conflict of interest.

## Acknowledgements

RB, CMM and ACM were supported by the EU H2020 CONSTRAIN project. We acknowledge the modelling groups who produced the CMIP6 simulations and the Earth System Grid Federation for providing data access. This study also benefited
from the ESPRI (Ensemble de Services Pour la Recherche à l'IPSL) computing and data center (https://mesocentre.ipsl.fr) which is supported by CNRS, Sorbonne Université, École Polytechnique and CNES and through national and international grants.

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
