# Peer review of "Model spread in multidecadal NAO variability connected to stratosphere-troposphere coupling"

_EGUsphere, 2023_

## Author Comment (AC1)

**Author response to reviews of manuscript "Model spread in multidecadal NAO variability connected to stratosphere-troposphere coupling"**

We thank the Editor for sourcing two thorough reviews of our manuscript. We are pleased both reviewers find merit in our study and support its publication in WCD subject to minor revisions. We have responded to the points raised below. We hope the Editor will find the revised manuscript improved and ready for publication.

Response to Reviewer 2

**The manuscript assesses multidecadal variability of the winter North Atlantic Oscillation (NAO) using historical simulations from 15 different climate models, each with at least 10 members. It is found that NAO variance is underestimated by models and potential causes are investigated. It is shown that NAO variance correlates significantly across the models with both the variance of the stratospheric polar vortex (SPV) and the coupling between the SPV and the NAO. Though causality cannot be identified, together these two factors explain 70% of the inter model spread in NAO variance. Furthermore, the coupling between the SPV and the NAO appears to be related to a measure of atmospheric eddy feedback. The authors also investigate other relationships finding no link to the Pacific, and a weaker though significant link to Atlantic Multidecadal Variability but only by processing in a way that precludes comparison with observations.**

**The manuscript is well written and the results are interesting. I recommend publication after addressing the minor comments below.**

First, we would like to thank the reviewer for her/his careful reading, interest in our study and the insightful comments that helped us to improve the manuscript. Figure S2 has been modified, the two missing models were added and we've changed the color code to match the figure on the article. Please find below the answers to the comments point-by-point. For clarity, all reviewer comments are in **bold** and responses in **blue**.

**Table 1: A minor point but I believe there are 50 ensemble members for MIROC6, or are the data you need not available?**

Indeed, not all members of the large ensembles provided daily wind data on the ESGF at the time of analysis. We added a sentence to clarify this in the revised manuscript:

"Therefore, we analyse CMIP6 models providing at least 10 ensemble members for the DECK historical experiment that provide daily zonal wind (ua) and meridional wind (va) variables needed to calculate the eddy feedback parameter described in Section 1 (see Section 2.2.5). Note that some models have additional ensemble members in ESGF which are not included here because they did not provide daily wind data at the time of analysis."

**Lines 143-144 Eddy feedback parameter: looks like something is missing here?**

Indeed, this was a problem with the .pdf file. We removed the "obj" that were in the PDF and we also added the unit hPa unit after 500 in the revised manuscript.

**Lines 158-159: please explain how serial correlation in the timeseries is accounted for**

As we do not test the significance of the correlation between variables (e.g. the AMV and NAO) within a single member, we do not need to take account for autocorrelation in the significance testing. All members are assumed to be independent and so the standard error on the ensemble mean calculated from the interensemble standard deviation follows the equation on L157. Each model is assumed to be independent for the purposes of the significance tests to assess intermodel relationships (L158-159).

**Fig 2: the model labels don't seem to match up with the dots - at least in my version**

Thanks. The labels are now fixed in the revised manuscript.

**Fig S2: looks like not all of the models are included here?**

Indeed, two models were missing and have now been added to the revised manuscript. The colors used have also been modified to match the rest of the manuscript.

**Fig 5a: is the ensemble mean the average of the regressions for the individual ensemble members, or is it the regression between the ensemble means of the NAO and AMV? I think it is the former (this needs clarifying) but the latter would be highlight forced responses and might also be interesting (though not so easy to compare with observations).**

In Figure 5a the ensemble mean is indeed the average of the regressions calculated for each individual member and not the regression between the ensemble means. We did not put the ensemble mean as it would not be directly comparable with the observations. It is now clarified in the legend of the Figure 5a:

"Figure 5: (a) Regression slope between the 20-year running mean NAO and AMV indices for DJFM over 1900-2010 for each model ensemble member (dot) and the ensemble mean of these regressions (diamond)."

**Fig 5b: if I am reading it correctly it looks like the strongest observed regressions occur with AMV leading NAO? If so, I wonder how that relates to previous work suggesting AMV can be explained by the integrated NAO e.g., https://journals.ametsoc.org/view/journals/clim/32/22/jcli-d-19-0177.1.xml**

This study and previous work by Li et al. (2013) relate low-pass filtered (7-year for O'Reilly et al., 2019 and 11-year for Li et al., 2013) AMV to the integrated NAO over all previous timesteps. This is different from our approach which uses 20-year running means for both AMV and NAO, so this could explain the apparent difference. Furthermore, O'Reilly et al. (2019) find around 60% of the AMV variance is not explained by the integrated NAO. In the manuscript we show the regression slope as a function of lead-lag which may not describe the statistical strength of the relationship. In Figure R9 we show the correlation coefficient of the NAO-AMV regression, which does reveal the strongest correlation with AMV leading the NAO by 10 years and weaker correlation with the NAO leading AMV consistent with the magnitude of the regression slopes in Figure 5b of the manuscript.

Meanwhile, both observational and modeling studies have shown that a positive AMV (associated with a stronger AMOC) can also induce a negative winter NAO response in the

atmosphere, as seen in Figure R9. Peing and Magnusdottir (2016), for example, showed that the positive phase of the AMV promotes negative NAO in winter using three different configurations of the Community Atmospheric Model version 5. Based on the long-term NOAA 20CRv3 (Slivinski et al., 2019) and ERA20C (Poli et al., 2016) reanalyses, Kwon et al. (2019) also showed the influence of the AMV on the NAO at multidecadal timescales.

[Figure]

**Figure R9:** *Lead-lag DJFM correlations between the 20-yr running mean AMV and the 20-yr running mean NAO for each model and the observations (dot) and their relative uncertainties (bars) calculated over the 1900-2014 period.*

**Fig S3: the caption is a bit confusing. Presumably this is the average of the values for each ensemble member, since the ensemble mean has been removed?**

Indeed, it is the average of the member values for each ensemble. We clarified the caption in the revised manuscript to avoid confusion:

"Figure S3: Scatter plot of the (a) average low frequency NAO variance (hPa²) and versus the regression slope calculated between the AMV and the NAO from the members for each of the ensemble of climate model simulations for the DJFM months over the 1850-2014 period. The ensemble mean is removed for both variables. The black line represents the least square regression with Pearson correlation r and p-value (see Section 2.3). (b) Same as (a) using a 10-yr running mean."

**Line 359: the values given in fig 9b appear to be different, r=0.52 p=0.05**

Sorry for this mistake, it is changed in the revised manuscript

**Discussion: I presume that constraining the NAO variance (lines 393-394) is beyond the scope of this study, though it would enhance it if it could be done.**

We considered using an emergent constraint methodology similar to Simpson et al. (2021), however, since the constrained NAO variance would still be systematically biased low wrt observations, this seems of limited value. We did not wish to mislead readers that a constrained distribution of NAO variability is more accurate given the evidence for other systematic biases.

**Discussion: Fig 5b appears to show further evidence of errors in all models. Perhaps a brief discussion of potential causes and implications could be included.**

Thanks for the suggestion. We have added to the sentence "This overall weak role for the AMV on the NAO may be related to weak atmosphere-ocean coupling in models as has been suggested in other studies (e.g., Simpson et al., 2018)." with "Indeed, the analysis in Figure 5b shows all models analysed fail to capture the lead-lag relationships between the NAO and AMV compared to observations. The amplitude of the NAO-AMV regression slope is too weak in the models with AMV leading the NAO, and has the opposite sign to observations when the NAO leads the AMV. Consistent with earlier studies (e.g. Kim et al. 2018; Simpson et al., 2018; Bracegirdle, 2022) this points to systematic errors in the modelled representation of atmosphere-ocean coupling in the North Atlantic on multidecadal timescales."

**References:**

Kwon, Y. O., Seo, H., Ummenhofer, C. C., & Joyce, T. M. (2020). Impact of multidecadal variability in Atlantic SST on winter atmospheric blocking. *Journal of Climate*, *33*(3), 867-892.

Li, J., Sun, C., & Jin, F. F. (2013). NAO implicated as a predictor of Northern Hemisphere mean temperature multidecadal variability. *Geophysical research letters*, *40*(20), 5497-5502.

Peings, Y., & Magnusdottir, G. (2016). Wintertime atmospheric response to Atlantic multidecadal variability: Effect of stratospheric representation and ocean–atmosphere coupling. *Climate dynamics*, *47*, 1029-1047.

Poli, P., Hersbach, H., Dee, D. P., Berrisford, P., Simmons, A. J., Vitart, F., ... & Fisher, M. (2016). ERA-20C: An atmospheric reanalysis of the twentieth century. *Journal of Climate*, *29*(11), 4083-4097.

Slivinski, L. C., Compo, G. P., Whitaker, J. S., Sardeshmukh, P. D., Giese, B. S., McColl, C., ... & Wyszyński, P. (2019). Towards a more reliable historical reanalysis: Improvements for version 3 of the Twentieth Century Reanalysis system. *Quarterly Journal of the Royal Meteorological Society*, *145*(724), 2876-2908.

---

## Author Comment (AC2)

**Author response to reviews of manuscript "Model spread in multidecadal NAO variability connected to stratosphere-troposphere coupling"**

We thank the Editor for sourcing two thorough reviews of our manuscript. We are pleased both reviewers find merit in our study and support its publication in WCD subject to minor revisions. We have responded to the points raised below. We hope the Editor will find the revised manuscript improved and ready for publication.

**Response to reviewer #1**

We would like to thank Dr Butler for her careful reading, interest in our study and the insightful comments that helped us to improve the manuscript. In the response we provide additional analysis to test the sensitivity of our results to regridding and the choice of the coupling parameter. In the manuscript, we have added observations to the NAO-IPV analysis. Figure S2 has been modified, the two missing models were added and we've changed the color coding to match the figures in the main article. Please find below the answers to the comments point-by-point. For clarity, all reviewer comments are in **bold** and responses in **blue**.

**General Comments**

**In this study, various factors that might explain the underestimation of multidecadal NAO variability in CMIP6 models are examined. The authors find that the representation of NAO-AMV coupling may explain some proportion of intermodel spread in multidecadal NAO variability, but that more of the spread is explained by the spread in multidecadal stratospheric polar vortex variability and stratosphere-troposphere coupling strength.**

**One overarching comment is that while some figures include a comparison to observed values (Figure 1-3, 5), particularly after the section on the AMV less is included about the observed relationships in relation to NAO-IPV and NAO-SPV. It's mentioned that the observed relationships are difficult to constrain, and in some cases it's clear that the value would fall well off the figure, but I think it might be worth including mention of the values in the text or captions (even if it's for a shorter period of time as in the case of the SPV), just to give some context for whether the models are even in the right ballpark. Or could you use the interannual values of the coupling for at least the NAO-SPV part to at least suggest what the relationship values might be for multidecadal timescales, as alluded to in the text and Figure S5?**

**Overall though this manuscript is well-written and the conclusions are well supported. The methods were explained clearly. The manuscript will be of interest to WCD readers. I only have minor suggestions.**

The observed relationship between the NAO and the IPV are now included in Figure S4 in the revised manuscript and are shown in Figure R1 below. The IPV multidecadal variance is now given in the legend of Figure 7. Because of missing values within the HadISST dataset in the Pacific over the period of interest, we only use the ERSSTv5 dataset.

Figure R1 shows a positive regression slope between the observed NAO and IPV at multidecadal scales, with only a few CMIP6 simulations reproducing a similar coupling. We

note that at interannual timescales there is a negative regression coefficient for the observations and a large part of the models (Figure R2). This opposite sign of coupling between the NAO and the IPV at multidecadal and interannual timescales is consistent with the recent study from Seabrook et al. (2023).

We added a point about this results in the revised manuscript (see below):

"A positive regression slope between the NAO and IPO is found within the observations, with only few simulations having a similar magnitude of relationship. As is the case for the AMV, this suggests that a bias is present in the models which could be related to atmosphere-ocean coupling or atmospheric teleconnections. We note that the NAO-IPO relationship on multi-decadal timescales has an opposite sign to that at interannual timescales (Figure S4), as found by Muller et al. (2008), and that the models and observations are in closer agreement at interannual timescales (not shown). This indicates that the bias in NAO connection with the tropical Pacific particularly appears at multidecadal timescales. Seabrook et al. (2023) hypothesised the opposite sign of the NAO-IPO relationship at multidecadal timescales is related to impacts of the IPV on stratospheric water vapour and subsequent impacts on the polar vortex. If models underestimated the stratospheric water vapour response it may explain the weak amplitude of the relationship. This is a topic for future study."

[Figure]

*Figure R1*: *Regression slope between the 20-year running mean NAO and IPV for DJFM over 1900-2010 for each model ensemble member (dot) and the ensemble mean (diamond). The NAO time series are detrended. The observed range of the regression slope (grey area) is defined as the minimum and the maximum of the slopes calculated from permutations of the observational datasets (ERSSTv5 for the IPV and 20CRv3, ERA20C and HadSLP2r for the NAO).*

[Figure]

*Figure R2: Regression slope between the interannual NAO and IPV for DJFM over 1900-2010 for each model ensemble member (dot) and the ensemble mean (diamond). The NAO time series are detrended. The observed range of the regression slope (grey area) is defined as the minimum and the maximum of the slopes calculated from permutations of the observational datasets (ERSSTv5 for the IPV and 20CRv3, ERA20C and HadSLP2r for the NAO).*

For the NAO-SPV analysis, we think it is best not to add observational estimates to the manuscript. As discussed in the paper, the record of stratospheric data is too short to robustly assess multidecadal variability. Even if we show that the stratosphere-troposphere coupling parameter on multidecadal timescales is correlated with the parameter on interannual timescales across models, the observed parameter still has considerable sampling uncertainty. We think this should be the subject of a specific study in order to investigate the potential of a constraint from the observations.

**Specific Comments**

**Line 53: By poor observational constraints, do you mean that the record is too short, or that the observations are poor in quality/high in uncertainty? I would specify here, even though it's explained more later in the paper.**

We do mean that the observational record is too short to robustly assess multidecadal variability. We clarified this point in the revised manuscript:

"However, the characteristics of multidecadal polar vortex variability in climate models are relatively understudied, in part because there are poor observational constraints due to the short record of stratospheric data."

**Line 85: It makes sense why the data were regridded for consistency; however was there any sensitivity testing done to ensure this does not significantly affect the results? It may be worth looking at one of the models with higher resolution and comparing how much the metrics change for the original vs regridded data.**

We tested the sensitivity of the results to regridding. The multidecadal NAO variance obtained using the native grid of each dataset or regridding the dataset to the CanESM5 grid (the coarsest model) are very similar (Figure R3) demonstrating this result is not sensitive to this pre-processing step. For the native and regridded datasets, there are also no differences in the relationship between multidecadal NAO variance and polar vortex variance, as well as between the multidecadal NAO variance and the regression slope between polar vortex strength and the NAO (Figure R4).

[Figure]

**Figure R3:** *(a) DJFM detrended 20-yr running mean NAO variance (hPa²) for each member of the 15 CMIP6 ensembles (dot), the ensemble mean (diamond) and the three observation-based datasets: 20CRv3 (solid line), ERA20C (dashed line) and HadSLP2r (dotted line) calculated over the 1900-2010 period. (b) Same but regridding all dataset based on the CanESM5 grid.*

[Figure]

**Figure R4:** (a) Scatter plot of the ensemble mean low frequency NAO variance versus (a) the low frequency polar vortex variance (m/s)$^2$ and (b) the regression slope between polar vortex strength and the NAO (hPa/ms$^{-1}$) for DJFM over 1900-2010. The black line represents the least square regression with Pearson correlation and p-value. (c) and (d) same as (a) and (b) but without regridding the data prior to the analysis.

Therefore, the paper results and conclusions are not sensitive to the regridding. We specify this point in the revised manuscript:

"All atmospheric data are regridded to the horizontal resolution of CanESM5, which is the coarsest model grid, using bilinear interpolation. The SST data are regridded over a regular 1°x1° grid using bilinear interpolation. We tested the sensitivity of the results to the regridding by recalculating the analysis using the native resolutions of datasets and find this does not affect the results shown in the paper."

**Line 119-120: Is this method preferable to say a linear fit of the data or some higher order fit? Did you test the sensitivity to other methods of trend removal?**

The global mean method is a common approach for detrending North Atlantic SST to compute the AMV index (Trenberth and Shea, 2006), as it attempts to remove global mean driven variability due to both external forcing and internal variability (e.g. due to ENSO). Since this is

an established method in the literature, and we wanted to be able to compare with other studies, we did not test other methods of trend removal.

**Line 130, line 139: Here do you mean "20-year running mean"? (is there a sliding window as described for the NAO in the caption of Figure 1?). Otherwise it sounds like one 20-year period but it's unclear whether they are overlapping or not.**

Indeed we meant "20-year running mean" instead of 20-yr mean. It is corrected in the revised manuscript.

**Line 136-139: Here you could just say "To identify SSWs, we use the index of […] between the months of December through March**

We modified this paragraph, which is now clearer in the revised manuscript:

"To identify SSWs, we use the index of Charlton and Polvani (2007) based on the temporary reversal of zonal mean zonal wind at 60°N and 10 hPa between the months of December through March. To be considered as discrete SSW events, periods of wind reversal to easterly must be separated by at least 20 consecutive days of westerly winds. We calculate the 20-year running mean winter SSW frequency and examine whether multidecadal variability in winter polar vortex strength is related to variability in SSW frequency (e.g., Jucker et al., 2014)."

**Line 222: Some of the models though seem like they could be overestimating AMV variability like EC-Earth3 (maybe a box and whiskers could be a way to identify where the obs value falls outside of the ~10th percentile of each model's ensemble distribution?).**

Indeed, some models like EC-Earth3 and the CNRM models could be overestimating AMV, although it is not possible to definitively conclude this since two EC-Earth3 members have lower than observed AMV variability. We have not added a box-whisker plot as the choice of percentile threshold would be somewhat arbitrary. Of course the further from the median the observations lie, the less likely it is they are consistent with the model, however, we cannot entirely rule out that the observations were just an unusual sample of climate variability. We added a point on the possible overestimation in the revised paper:

"Some models are characterized by larger ensemble mean AMV variability and larger ensemble spread which encompasses the observations (CMCC-CM2-SR5, CNRM-CM6-1, CNRM-ESM2-1, EC-Earth3 and IPSL-CM6A-LR), potentially overestimating the variability (e.g. EC-Earth3), while others have very weak average AMV variability and smaller ensemble spread (INM-CM5-0, MIROC-ES2L, MIROC6, MPI-ESM1-2-LR)."

**Line 225: Were the relationships between the NAO variance and some measure of the periodicity of the AMV (Fig S2) considered, in a similar manner to Figure 4?**

To test this suggestion, we estimate the AMV periodicity by selecting the period corresponding to the maximum AMV power for each model (Figure S2 from Supplementary Information), considering only periods between 150 and 10 years as we are interested in multidecadal variability. There is a non-significant positive relationship between the periodicity of the AMV at the corresponding maximum power spectra and the variance of the NAO (Figure R5). Although other investigations could be made, this first analysis suggests that the multidecadal

NAO variance spread doesn't relate to the periodicity of the AMV within CMIP6 models. However, since we use a fixed averaging window for the NAO of 20 years, it is not clear how spread in the periodicity of the AMV would affect the 20-year running mean NAO, instead the amplitude of AMV variability on the same timescale seems more relevant, so we keep only this in the manuscript.

[Figure]

**Figure R5:** Scatter plot of the ensemble mean 20-year running mean NAO variance (hPa²) versus the AMV periodicity (year) for the maximum power spectra (Figure S2 from supplementary information) considering periods between 10 and 150 years.

**Line 234: For coupling metrics like the NAO-AMV coupling parameter and later the stratosphere-troposphere coupling parameter, I wonder how sensitive these results are to using correlations instead of regression coefficients? They should be similar of course, but the regression is also related to the spread of the AMV variance (in this case) so the correlation might be simpler to interpret in some ways.**

We tested the sensitivity of the NAO-AMV coupling and the stratosphere-troposphere coupling to using correlations instead of regression coefficients (see below). The lead-lag correlations between the AMV and NAO are consistent with results obtained using regression coefficients, with a large proportion of models showing negative correlation in line with observations, but strongly underestimating it (Figure R6). The strong negative correlation found at negative lags in the observations is consistent with previous studies (Peings et al. 2016; Kwon et al. 2020). There is no significant relationship between the NAO variance and the NAO-AMV correlation, consistent with the result using the regression coefficient (Figure R7). Finally, a similar positive significant relationship between the NAO variance and the NAO-vortex coupling is found using correlations instead of the regression coefficients (Figure R8). We specify that the results are similar for the correlation coefficient as for the regression slope in the revised manuscript.

[Figure]

**Figure R6:** *Lead-lag DJFM correlations between the 20-yr running mean AMV and the 20-yr running mean NAO for each model and the observations (dot) and their relative uncertainties (bars) calculated over the 1900-2014 period.*

[Figure]

**Figure R7:** *Scatter plot of the ensemble mean 20-year running mean NAO variance (hPa²) versus the correlation between the 20-year running mean NAO and AMV for DJFM over the 1900-2010. The AMV leads the NAO by 10 years.*

[Figure]

**Figure R8:** *Scatter plot of the ensemble mean 20-year running mean NAO variance (hPa²) versus the correlation between polar vortex strength and the NAO for DJFM over 1900-2010. The black line represents the least square regression with Pearson correlation and p-value.*

**Line 292-293: The sensitivity of the results could also be true though for other results, such as Figure 7b where the removal of the MIROC models (or the two "worst" models in each case) might result in better/worse relationships. I guess I'm not sure it's "fair" to point that out here only for this part of the paper?**

This is a fair point. We have removed the sentences on L292-293.

**Figure 8b: I didn't find where this panel is described in the text other than line 295, however that just refers to the stratosphere-troposphere coupling parameter, not the relationship between this parameter and the NAO variance.**

Indeed, the relationship between the stratosphere-troposphere coupling parameter and the NAO variance wasn't properly described. It is now added in the revised manuscript:

"A strongly significant positive relationship is found between the low-frequency NAO variability and the stratosphere-troposphere coupling parameter in the models (Figure 8b). A similar result is found using correlations instead of regression coefficients (not shown)."

**Line 334-335: This result is a little counterintuitive, given what is said on line 229. Could you explain more? Does it have to do with these relationships not necessarily explaining the spread across models even though they may explain physical relationships within a single model?**

Figure R9 shows there is a relationship between the climatological vortex strength and SSW frequency across models, as has been noted elsewhere (e.g., Wu and Reichler, 2020). The causality of this relationship has been proposed as the climatological winds setting the conditions for wave propagation and subsequent SSW onset. However, this relationship is not

consistently found across ensemble members within single models (Figure R10), indicating that it may not be as relevant for understanding internal variability in SSW frequency. This is in agreement with the figure in the main text which shows that differences in internally-generated multidecadal vortex *variability* are not related to SSW *variability*. Causality is difficult to establish in this framework because, as previously mentioned, studies have identified that vortex strength plays a role in setting the conditions for SSWs but one may also expect SSWs to *contribute to* vortex variability. Nevertheless, the absence of relationship suggests both points are not supported by the analysis and multi-decadal vortex variability must be driven by other processes.

[Figure]

Figure R9: Scatter plot of the ensemble mean of the number of SSW versus the 20-yr running mean detrended polar vortex strength (m/s) for 1900-2010 and for DJFM.

[Figure]

Figure R10: 20-year running mean DJFM vortex strength vs. number of SSWs for 1900-2010 and for individual ensemble members per CMIP6 model used in the main manuscript. The vortex strength is detrended before the analysis.

We have amended the manuscript as follows:

"No significant relationship is found between the variability of 20-year running mean SSW frequency and the low frequency polar vortex variability over 1900-2010 (Figure 9a). The lack of relationship could potentially be related to differences in the amplitude of SSWs in models, and therefore the role of SSWs in *driving* polar vortex variability would differ. It may also be because low frequency polar vortex variability is driven by processes unrelated to SSWs, such as low frequency variability in the upward propagation of planetary wave activity (e.g., Schimanke et al. 2011) or that model biases in subtropical lower stratospheric wind speeds affect the sensitivity of the vortex to upward propagating waves (Sigmond and Scinocca, 2010), but this is beyond the scope of the current study. However, we note that causality is difficult to establish in this framework as the vortex strength plays a role in setting the conditions for SSWs to occur (Hall et al., 2021; Wu and Reichler, 2020)."

**Line 336-337: Alternatively, what about the climatological background wind speeds in the subtropics, as proposed by e.g., Sigmond and Scinocca 2010?**

Sentence amended to: "It may also be because low frequency polar vortex variability is driven by processes unrelated to SSWs, such as low frequency variability in the upward propagation of planetary wave activity (e.g., Schimanke et al. 2011) or that model biases in subtropical lower stratospheric wind speeds affect the sensitivity of the vortex to upward propagating waves (Sigmond and Scinocca, 2010), but this is beyond the scope of the current study."

**Technical Corrections**

Thank you for these corrections, modifications made.

**Lines 143-144: There were some issues in my pdf rendering at least with symbols in these sentences (there are weird missing "obj" symbols appearing around the references). Also there are no units after "500"**

Done

**Line 188: add "multidecadal" in front of "NAO"**

Done

**Line 190: change "seems to be" to "is"**

Done

**Figure S1: x-axis labels are shifted off the tick marks**

Fixed

**Figure 2: Two comments. The first is that the x-axis labels are shifted oddly here and don't line up with the tick marks. The second is that at least by eye, the lines for 20CRv3 and ERA20C seem to lie higher than the halfway point between 1.0-1.5 tickmarks on the y-axis, but according to Figure 1c they should both lie at 1.15.**

Indeed, there was a difference in the observations between Figures 1b and 2, which came from an error in Figure 2, which used the 20CRv3 and ERA20C on their native grids instead of regridded to the CanESM5 grid as in Figure 1b. We changed Figure 2 to be consistent with Figure 1b and rearranged the x-axis in the revised manuscript.

**Figure S2: here the time period of 1900-2014 is mentioned though most of the other figures use 1900-2010. Is it possible to include the spectra for the observed datasets here? Or not really since these are ensemble-mean estimates of the forced response?**

Sorry for the mistake, it is now changed in the revised manuscript. Indeed, we did not add the observations on this plot as these are ensemble-mean estimates of the forced response which cannot be derived for observations. Moreover, the power spectrum is calculated for each model

based on all the members available, whereas it would be only calculated over one realisation (and so limited to 110 years) for the observations, which could influence the results.

**Line 206: change to "simulated AMV variance."**

Done

**Line 237: change "is toward the lower end" to "on the more negative end"**

Done

**Line 333: remove url link**

Done

**Figure 9a: typo in x-axis label**

Done

**Line 368: remove "do"**

Done

**References:**

Peings, Y., Simpkins, G., & Magnusdottir, G. (2016). Multidecadal fluctuations of the North Atlantic Ocean and feedback on the winter climate in CMIP5 control simulations. *Journal of Geophysical Research: Atmospheres, 121*(6), 2571-2592.

Seabrook, M., Smith, D. M., Dunstone, N. J., Eade, R., Hermanson, L., Scaife, A. A., & Hardiman, S. C. (2023). Opposite impacts of interannual and decadal Pacific variability in the extratropics. *Geophysical Research Letters*, *50*(2), e2022GL101226.

Simpson, I. R., K. A. McKinnon, F. V. Davenport, M. Tingley, F. Lehner, A. Al Fahad, and D. Chen, 2021: Emergent Constraints on the Large-Scale Atmospheric Circulation and Regional Hydroclimate: Do They Still Work in CMIP6 and How Much Can They Actually Constrain the Future? *J. Climate*, **34**, 6355–6377, https://doi.org/10.1175/JCLI-D-21-0055.1.

Wu, Z., & Reichler, T. (2020). Variations in the frequency of stratospheric sudden warmings in CMIP5 and CMIP6 and possible causes. *Journal of Climate*, *33*(23), 10305-10320.